

# A proposed update for the classification and description of bacterial lipolytic enzymes

Thomas C.A. Hitch and Thomas Clavel

Functional Microbiome Research Group, Institute of Medical Microbiology, University Hospital of RWTH Aachen, Aachen, Germany

## ABSTRACT

Bacterial lipolytic enzymes represent an important class of proteins: they provide their host species with access to additional resources and have multiple applications within the biotechnology sector. Since the formalisation of lipolytic enzymes into families and subfamilies, advances in molecular biology have led to the discovery of lipolytic enzymes unable to be classified via the existing system. Utilising sequence-based comparison methods, we have integrated these novel families within the classification system so that it now consists of 35 families and 11 true lipase subfamilies. Representative sequences for each family and subfamily have been defined as well as methodology for accurate comparison of novel sequences against the reference proteins, facilitating the future assignment of novel proteins. Both the code and protein sequences required for integration of additional families are available at: https://github.com/thh32/Lipase_reclassification.

## INTRODUCTION

Bacterial lipolytic enzymes are involved in a diverse range of microbial functions from metabolism of fat for energy (*Drouault et al., 2002*) to pathogenesis (*Canaan et al., 2004*). These enzymes are also of interest to the biotechnology industry for production of biodiesel and biopolymers (*Jaeger & Eggert, 2002*). The alpha-beta hydrolase family contains both lipases (Enzyme commission (EC): 3.1.1.3) and carboxylesterases (EC: 3.1.1.1), both of which can catalyse the hydrolysis of 4-nitrophenyl-bound fatty acids (*Guo et al., 2010*; *Rao et al., 2011*). Lipases are defined as those which display some ability to catalyse the release of free fatty acids from long-chain triacylglycerols, such as those found in plant oil and animal fat, however activity may be highest against short or medium chain fatty acids (*Jaeger & Eggert, 2002*; *Levisson, Van der Oost & Kengen, 2007*; *Glogauer et al., 2011*). This separates the lipases from the esterases, which may have activity against p-nitrophenyl (pNP)-bound fatty acids (a standard test for substrate specificity) but are unable to act upon triacylglycerol molecules (*Yu et al., 2010*; *Masomian et al., 2016*). A further subset of lipases are the true lipases which are defined as lipases which display preferential activity against long-chain fatty acids (>C10) (*Sorokin & Jones, 2009*). The gold standard classification system of bacterial lipolytic enzymes was originally published by *Arpigny & Jaeger (1999)*

Corresponding author
Thomas C.A. Hitch,
thitch@ukaachen.de,
th.tomhitch@gmail.com

and defined eight families based on the biochemical properties and sequence similarity of 53 proteins.

Under the *Arpigny & Jaeger (1999)* classification system, true lipases were placed within a singular family (family I), within which sub-families are defined by unique lipase groups (*Arpigny & Jaeger, 1999*; *Jaeger & Eggert, 2002*). Originally, six sub-families of true lipases were described based on shared features such as source genus, requirement of a chaperone for folding, motifs and sequence similarity (*Arpigny & Jaeger, 1999*). *Jaeger & Eggert (2002)* redefined these families by splitting family I.5 into two new families, family I.5 and family I.6, and moving the previous members of family I.6 into family I.7 (*Jaeger & Eggert, 2002*). In the 20 years since this bacterial lipolytic enzyme classification system was published, more than twice as many 'novel' families have been proposed (*Handrick et al., 2001*; *Ewis, Abdelal & Lu, 2004*; *Lee et al., 2010*; *Lee et al., 2006*; *Levisson, Van der Oost & Kengen, 2007*; *Bender et al., 2009*; *Kim et al., 2009*; *Hu et al., 2010*; *Rao et al., 2011*; *Jeon et al., 2011*; *Bassegoda, Pastor & Diaz, 2012*; *Charbonneau & Beauregard, 2013*; *Zarafeta et al., 2016*; *Castilla et al., 2017*; *Parapouli et al., 2018*). A recent update to this system expanded the system to include some (11 of 30) of the recently published novel lipases within 19 families and eight true lipase sub-families, however many lipolytic proteins remain unclassified (*Kovacic et al., 2019*).

Many claims of identifying novel families of lipolytic enzymes have been made, however no exhaustive comparison has been conducted to verify and combine these results. This has led to an inconsistent characterisation system and multiple publications claiming discovery of the same family (*Parapouli et al., 2018*). Examples of this are family X (*Levisson, Van der Oost & Kengen, 2007*; *Bassegoda, Pastor & Diaz, 2012*) and family XV (*Charbonneau & Beauregard, 2013*; *Zarafeta et al., 2016*), both of which have been coined twice. *Rao et al. (2013)* attempted to incorporate previously published families into their analysis, however their literature search did not include multiple families (*Chu et al., 2008*; *Bender et al., 2009*; *Hu et al., 2010*; *Lee et al., 2010*; *Jeon et al., 2011*; *Nacke et al., 2011*; *Bassegoda, Pastor & Diaz, 2012*). *Ferrer et al. (2015)* also attempted to integrate additional families into the *Arpigny & Jaeger (1999)* classification system. Both previous attempts at including additional families relied on the authors claims of 'novelty' without reanalysis of these claims and, as discussed above, incomplete literature searches mean additional 'novel' families have been missed from the analysis. The possibility exists that multiple publications may describe members of the same 'novel' family under different names, artificially inflating the number of families. Therefore, before additional families can be incorporated into the system and provided with a family number, previous claims of novelty must be confirmed and integration of these proteins into a singular classification system must occur.

Alternative classification systems for bacterial lipolytic enzymes have been published alongside online databases dedicated to these new classification systems, including MELDB (*Kang et al., 2006*) and LIPABASE (*Messaoudi et al., 2011*). Both websites have since been deactivated preventing their continued use. The simplicity of the *Arpigny & Jaeger (1999)* classification system is that by providing type proteins for each family, the system can continue to be utilised by the research community. Additional databases such as the Lipase Engineering Database (*Pleiss et al., 2000*) and ESTHER (*Lenfant et al., 2013*) classify similar

groups of enzymes, however these are not specific to bacterial lipolytic enzymes and as with MELDB and LIPABASE, their continued availability is not guaranteed.

In this paper, the novelty of lipolytic enzymes was confirmed via pairwise sequence comparison to the current type proteins. This allowed us to expand the current classification system and remove spurious claims of novelty and provide a consistent nomenclature to these proteins.

## METHODS

### Literature search

Multiple methods were used to identify post-1999 claims of novel lipases and lipase families. The key terms 'novel, bacterial, lipase, esterase, carboxylesterase, family' were used to search both the Pubmed (http://www.ncbi.nlm.nih.gov/pubmed/) and Google scholar (https://scholar.google.de/) databases. Each identified paper was screened to confirm that: (i) the protein was of bacterial/metagenomic origin; (ii) the lipolytic activity of the proposed sequence was confirmed experimentally; (iii) the studied lipolytic protein was indeed novel or assigned to a previously defined family.

### Sequence comparison

Type proteins for both the families described by *Arpigny & Jaeger (1999)* and lipolytic proteins identified by the literature search were compared pairwise via ClustalW (*Thompson, Higgins & Gibson, 1994*). Sequence identity was calculated using a python implementation of the ''Identity and similarity'' tool within the Sequence Manipulation Suite (*Stothard, 2000*) and has been made available as a reproducible Jupyter notebook (Pairwise_analysis.ipynb) within the GitHub repository given in the abstract. Proteins were considered likely to be part of the same family at 60% sequence identity, a threshold that is three standard deviations away from the mean similarity between the type proteins of the original eight families ($30.9 \pm 9.6$% similarity). Proteins identified by sequence comparison to be representatives of the same family had their biochemical properties compared to identify if these were consistent or highly divergent. Transmembrane regions were detected within each protein using TMHMM (*Krogh et al., 2001*). Proteins were also annotated with an ESTHER family using the online HMMscan tool, minimum bitscore of 50 (*Lenfant et al., 2013*), allowing for cross-classification system comparison.

### Taxonomic occurrence

To provide insight into the evolutionary diversity or specificity of each family, the NCBI genome database (downloaded November 2017) (*Benson et al., 2008*) containing 72,462 genomes was annotated against the type proteins for each family using BLASTP (*Altschul et al., 1990*). Matches were separated into two categories: (i) strict matches required high-scoring-pairs to share >80% identity across >80% of both the query and subject sequences, providing an overall identity of >64%. Minimum identity of 60% was selected based on previous research on functional annotation transfer (*Radivojac et al., 2013*); (ii) An expanded genome search using reduced strictness (>60% identity across >80% of the query sequence, resulting in a total identity of >48%) was conducted to provide insight

into the potential host range of metagenome-derived proteins with no close matches. The coverage and identity used in the expanded genome search remain more stringent than those previously identified to be optimum for protein family identification and clustering (*Veeramachaneni & Makałowski, 2004*). Once annotated, genomes were linked to their taxonomic lineage using the NCBI taxonomy information, implemented in MGkit (*Rubino et al., 2014*), and the Genome Taxonomy database (GTDB), version 86 (*Parks et al., 2018*). GTDB annotations were used to confirm the taxonomic assignment of genomes when the NCBI assignment was uncertain.

## Origin check of metagenomic enzymes

Metagenomic approaches capture both eukaryotic and prokaryotic DNA. To prevent the inclusion of eukaryotic enzymes into this system, enzymes captured by metagenomic approaches were screened against the NCBI genome database. If a strict match was identified, the bacterial origin of the enzyme was assured. For those enzymes without a genome match, possibly due to the host species being as-yet uncultured, the sequences were compared against the NR database using BLASTP. The taxonomic assignment of the top 10 matches was examined and if all were bacterial, the enzyme was included within the classification system, else they were excluded.

# RESULTS AND DISCUSSION

## Overview

Pairwise sequence comparison of the accepted type proteins from *Arpigny & Jaeger (1999)* to each proposed new family retrieved from the literature identified a single match (>60% similarity), Table S1. The HZ lipase proposed by *Masomian et al. (2016)* shared 69.5% similarity with family I.5 (U78785) and 57.0% similarity with family I.6 (M1271) (*Masomian et al., 2016*). Due to this, the HZ lipase was assigned as a member of family I.5. Choi et al. (2004) also proposed a novel lipase (EstI) that was identified to share 62.5% similarity with family I.6 (M12715), which indicates it is a member of this subfamily.

Pairwise comparison of the proposed families type proteins against one-another was conducted to identify if any families have been discovered multiple times (Table S2). We identified that Est30 (*Ewis, Abdelal & Lu, 2004*), EstUT1 (*Samoylova et al., 2018*) and EstOF4 (*Rao et al., 2013*) share >60% similarity, forming a single family, as did EstGtA2 (*Charbonneau, Meddeb-Mouelhi & Beauregard, 2010*), bMGL (*Kitaura, Suzuki & Imamura, 2001*) and LipS (*Chow et al., 2012*).

As this paper aims to act as an expand upon the existing *Arpigny & Jaeger (1999)* classification system which was updated very recently (*Kovacic et al., 2019*), all proposed families were included within the existing framework. We remained consistent with the recent update to the original classification system proposed by *Kovacic et al. (2019)*, however we propose a few corrections. In the aforementioned update, LipG was identified as the type protein for family XI, however its preference for long-chain fatty acids (C14-16) and activity against triglycerides (tricaprylin) identify it as a true lipase (*Lee et al., 2006*). Secondly, EstD and LipR were both placed together under family X, however these proteins share <30.1% similarity and have different biochemical characteristics, suggesting they

represent separate families. This is supported by each proteins catalytic serine being contained within different motifs. Due to this we propose dividing family X into family X and family XXVIII according the numbering proposed in the present study. Finally, in the updated classification system by Kovacic et al. family XVIII was based on EstUT1, separate from Est30 and EstOF4 against which it shares however high (>60%) similarity. Due to this we have collapsed family XVIII into family XIII. These changes mean that family I to XIX remain consistent with the published updated classification system, apart from family XI and XVIII. To prevent confusion regarding our changes up to family XIX we have provided descriptions for each family starting at family IX (see Table 1 and individual descriptions provided below).

## Proposed family I subfamilies
### Family I, subfamily 9
LipG, the type protein for family I.9, was isolated during a metagenomic study of korean tidal flat sediment (*Lee et al., 2006*). The GxSxG motif (where x represents any amino acid) was identified as GHSLG. Genome analysis was unable to find similar proteins within the current genome database (Table S3). However, sequence matches within the NR database confirmed its bacterial origin (Table S4). Maximum activity was identified against C14 and C16, however activity decreased by 65% against C18. Both $Ca^{2+}$ and $Mn^{2+}$ were identified to increase the activity of LipG and chelation of divalent metal ions via ethylenediaminetetraacetic acid (EDTA) significantly reduced activity confirming the necessity of metal ions. The effect of temperature and pH have yet to be reported for LipG. LipG was annotated as belonging to 'Lipase_3' under the ESTHER system.

### Family I, subfamily 10
RlipE1 was isolated from the rumen of Chinese Holstein cattle using a metagenomic bacterial artificial chromosome (BAC) library (*Liu et al., 2009*). Whilst no genomes contained similar proteins in our analysis, the closest matches within the NR database were bacterial in origin, indicating that RlipE1 originated from bacteria. The catalytic Serine was identified within GLSMG. Activity against triglycerides along with maximum activity against C16 confirmed that RlipE1 is a true lipase. Optimum conditions were identified as 30 °C and pH 7-8. The effect of metal ions has yet to be examined. RlipE1 was annotated as 'A85-Feruloyl-Esterase' under the ESTHER system.

### Family I, subfamily 11
MPlaG is a phospholipase isolated from tidal flat sediment in korea (*Lee et al., 2012*). The catalytic serine is present within the pentapeptide GHSLG. Activity against triglycerides and a preference for long-chain fatty acids identified MPlaG as a true lipase. Optimum conditions were determined as pH 8 and 25 °C. Addition of $Ca^{2+}$ increased activity 10-fold whilst $Cu^{2+}$, $Zn^{2+}$, and $Ni^{2+}$ all decreased activity. EDTA (5 mM) removed all activity, suggesting $Ca^{2+}$ is essential for activity. Expanded genomic analysis identified possible members of this enzyme family within the *Oceanospirillaceae*. The source of isolation of MPlaG along with its potential host identified by genome analysis suggest that family I.11 may be specific to the marine environment. MPlaG had no significant match to a ESTHER family.

**Table 1  Bacterial lipolytic enzyme families.** Table summerising each bacterial lipolytic family. Each family is provided with a type protein for which the accession number and name is provided along with its proposed function. The original publication in which each family was integrated into the classification system is stated.

| Family | Sub-family | Type protein | Publication | Accession | Function |
|---|---|---|---|---|---|
| **Family I** | 1 | | *Arpigny & Jaeger (1999)* | D50587 | True lipase |
| | 2 | | *Arpigny & Jaeger (1999)* | X70354 | True lipase |
| | 3 | | *Arpigny & Jaeger (1999)* | D11455 | True lipase |
| | 4 | | *Arpigny & Jaeger (1999)* | M74010 | True lipase |
| | 5 | | *Arpigny & Jaeger (1999)* | U78785 | True lipase |
| | 6 | | *Jaeger & Eggert (2002)* | M12715 | True lipase / Phospholipase |
| | 7 | | *Jaeger & Eggert (2002)* | X99255 | True lipase |
| | 8 | Lip1 | This paper | Q3IF07 | True lipase |
| | 9 | LipG | This paper | DQ458963 | True lipase |
| | 10 | RlipE1 | This paper | FJ529693 | – |
| | 11 | MPlaG | This paper | EU285670 | True lipase / Phospholipase |
| **Family II** | | | *Arpigny & Jaeger (1999)* | P10480 | Secreted acyltranferase |
| **Family III** | | | *Arpigny & Jaeger (1999)* | M86351 | Extracellular lipase |
| **Family IV** | | | *Arpigny & Jaeger (1999)* | X62835 | Esterase |
| **Family V** | | | *Arpigny & Jaeger (1999)* | M58445 | PHA-depolymerase |
| **Family VI** | | | *Arpigny & Jaeger (1999)* | D90904 | Carboxylesterases |
| **Family VII** | | | *Arpigny & Jaeger (1999)* | Q01470 | Carbamate hydrolase |
| **Family VIII** | | | *Arpigny & Jaeger (1999)* | AAA99492 | Stereoselective esterase |
| **Family IX** | | phaZ7 | *Kovacic et al. (2019)* | AY026355 | Poly(3-hydroxylbutyrate) depolymerase |
| **Family X** | | EstD | This paper | NP_228147 | Carboxylesterase |
| **Family XI** | | EstA | This paper | ABY60416 | Esterase |
| **Family XII** | | LipEH166 | | EU515239 | Lipase |
| **Family XIII** | | Est30 | This paper | AY186197 | Carboxylesterase |
| **Family XIV** | | EstA3 | *Kovacic et al. (2019)* | NP_623858 | Carboxylesterase |
| **Family XV** | | EstGtA2 | *Kovacic et al. (2019)* | AEN92268 | Carboxylesterase |
| **Family XVI** | | LipSM54 | *Kovacic et al. (2019)* | AGF29555 | Lipase |
| **Family XVII** | | LipJ2 | *Kovacic et al. (2019)* | KX096709 | Lipase |
| **Family XVIII** | | plaB | This paper | EF408871 | Phosphatidylcholine-specific phospholipase |
| **Family XIX** | | LipSM | *Kovacic et al. (2019)* | KX353755 | Lipase |
| **Family XX** | | FLS18D | This paper | ACL67852 | Esterase |
| **Family XXI** | | FnL | This paper | ABS61180 | Lipase |
| **Family XXII** | | EstD2 | This paper | GQ866023 | Secreted esterase |
| **Family XXIII** | | LipA | This paper | ACJ13070 | Lipase |
| **Family XXIV** | | Rv0045c | This paper | NP_214559 | Esterase |
| **Family XXV** | | EM3L4 | This paper | GQ340926 | Lipase |
| **Family XXVI** | | EstGS | This paper | HQ156900 | Secreted esterase |
| **Family XXVII** | | EstGH | This paper | HQ156914 | Esterase |
| **Family XXVIII** | | LipR | This paper | YP_345621 | Lipase |

**Table 1** (*continued*)

| Family | Sub-family | Type protein | Publication | Accession | Function |
|---|---|---|---|---|---|
| **Family XXIX** | | EstGX1 | This paper | HQ262952 | Lipase |
| **Family XXX** | | EstP2K | This paper | JN001203 | Esterase |
| **Family XXXI** | | Est12 | This paper | KF313138 | Esterase |
| **Family XXXII** | | Est9x | This paper | AFR79233 | Lipase |
| **Family XXXIII** | | EstDZ2 | This paper | KX301277 | Secreted carboxylesterase |
| **Family XXXIV** | | Est22 | This paper | GAC12013 | Homoserine transacetylase |
| **Family XXXV** | | AcXE2 | This paper | ACM59679 | Acetyl xylan esterase |

## Proposed families
### Family X

EstD (44.5 kDa) was isolated from *Thermotoga maritime* (*Levisson, Van der Oost & Kengen, 2007*), a hyperthermophile isolated from geothermally heated marine sediment (*Latif et al., 2013*). EstD contains the GHSLG variant of the GxSxG motif. Activity was reduced by 96% when treated with PMSF. Both highly similar and reduced threshold matches to EstD were specific to the *Thermotoga* genus. EstD activity increased linearly with temperature to a maximum of 100 °C at which it has a half life of 1 h. The optimum pH of EstD was 7 with a maximum activity against C5 to C8. EstD activity is not significantly affected by the presence or absence of metal ions or by treatment with EDTA, suggesting metal ions may have a minor effect but are not essential for activity. EstD was annotated as 'Bacterial_EstLip_FamX' under the ESTHER system.

### Family XI

The type protein for family XI, EstA (33 kDa) was identified via lipase screening of a bacterial artificial chromosome (BAC) library created using DNA fragments from the South China Sea (*Chu et al., 2008*). Analysis of EstA and its closest relatives identified the GxSxG variant, GHSMG, within all members. PMSF reduced activity of EstA by 97%. Strict genome analysis was unable to identify any matching sequences but the expanded genome search identified matches were found within 16 taxonomic families and 41 genera. These results suggest that while the exact protein is likely to be present in a uncultured species, variants exist within a diverse range of hosts. EstA is specific for short- and medium-chain fatty acids with maximum activity against C6, decreasing till C10 past which no activity is detected. The optimum temperature and pH for EstA was determined to be 45 °C and 6.5. The metal ions $Mg^{2+}$, $Mn^{2+}$ and $Co^{2+}$ increase EstA activity while $Fe^{2+}$, $Fe^{3+}$ and $Zn^{2+}$ reduced activity. The effect of EDTA was not tested. EstA was annotated as belonging to the 'A85-EsteraseD-FGH' family within the ESTHER classification system.

### Family XIII

*Ewis, Abdelal & Lu (2004)* identified Est30 (30 kDa), the type protein for family XIII, within *Geobacillus stearothermophilus* ATCC7954 (*Ewis, Abdelal & Lu, 2004*). EstUT1, identified by *Samoylova et al. (2018)*, was determined to be the second member of this family due to sharing 65% sequence similarity and key biochemical characteristics (*Samoylova et al., 2018*). EstOF4 was also identified to be a member of this family based on a protein

similarity of 85.6% and 60.9% to Est30 and EstUT1 respectively (*Rao et al., 2013*). The GxSxG motif was identified within all members as GxSLG and treatment with PMSF inhibited enzymatic activity. Optimum conditions for family XIII are 50–80 °C and pH 8-9. All members were determined to be most active against short and medium-chain fatty acids and display no activity against long-chain fatty acids. EDTA had no effect on either lipases, suggesting activity is independent of metal ions. Strict genome analysis identified the presence of proteins similar to Est30 within four genera (*Anoxybacillus, Bacillus, Geobacillus, Parageobacillus*) belonging to the *Bacillaceae*, in which both EstOF4 and EstUT1 was also identified. Expanded genome analysis identified matches in five taxonomic families and 31 genera according to the NCBI taxonomy. This suggests that members of this enzymatic family are present within a wide range of taxonomic groups. All three members of this family were assigned to 'CarbLipBact_1' under the ESTHER classification system.

### Family XVIII

PlaB, a phospholipase isolated from *Legionella pneumophila*, is the type protein for family XVIII (*Bender et al., 2009*). Family XVIII contains its catalytic triad serine within the motif THSTG, replacing the first glycine in the common GxSxG motif with threonine. Genome analysis identified family XVIII as monophyletic to the *Legionellaceae*, specifically the *Legionella* and *Fluoribacter* genera. Due to the clinical relevance of *Legionella pneumophila*, research on PlaB has focused on its possible role as a virulence factor via hydrolysis of phosphatidylcholine within eukaryotic membranes. PlaB was annotated to its own ESTHER family called 'PlaB'.

### Family XIX

Cultivation of lipase positive isolates from waste-treatment sludge, determined by Rhodamine B-olive oil plates, identified *Stenotrophomonas maltophila* Psi-1 and the identification of LipSm (40.7 kDa) (*Parapouli et al., 2018*). LipSm activity was tested against two pNP-bound fatty acids (C4 and C12) with maximum activity against C12. Optimum activity was achieved at 30–40 °C and pH 7 and all tested metal ions ($Ca^{2+}$, $Mg^{2+}$, $Mn^{2+}$, $Na^{2+}$) inhibited activity. Sequence analysis identified the presence of the catalytic serine within the pentapeptide GHSQG and similar proteins only in the *Stenotrophomonas*. The ability of purified LipSm to release free fatty acids from olive oil was not tested, preventing its classification as a true lipase. LipSm was annotated as belonging to the 'Fungal-Bact_LIP' family under the ESTHER classification system.

### Family XX

*Hu et al. (2010)* isolated two lipolytic enzymes sharing identity with a putative poly(3-hydroxybutyrate) depolymerase, FLS18C and FLS18D (∼32 kDa), from the South China Sea forming the novel family XX (*Hu et al., 2010*). As biochemical characterisation has only been conducted for FLS18D, it was selected as the type protein. The catalytic serine was identified within the conserved GHSMG motif. Activity was highest against C4, with decreasing activity until C10, suggesting preferential hydrolysis of short-chain fatty acids. Optimum activity was achieved at 45 °C at pH 8. The effect of metal ions on the activity of

FLS18D has not yet been studied. Genome analysis was unable to identify similar proteins within any sequenced bacteria but bacterial origin was identified via BLAST annotation against the NR database. Both proteins within this family were annotated as members of the 'Abhydrolase_5' family within ESTHER.

### Family XXI

*Yu et al. (2010)* identified the type protein, FnL (∼30 kDa), for family XXI within *Fervidobacterium nodosum Rt17-B1*, an anaerobic thermophile with optimum growth at 80 °C (*Yu et al., 2010*). Genome analysis identified FnL as specific to the *Fervidobacterium*. The pentapeptide sequence AxSxG identified within FnL and its closest relatives in the original analysis. Treatment with PMSF reduced activity to 5%, confirming FnL as a serine hydrolase. FnL showed the highest activity towards medium length fatty acids, with highest activity on C10. Whilst initially published as a subfamily of true lipases, its preference to medium-chain fatty acids identifies it as a novel family of non-true lipases. FnL showed the highest activity at 70 °C and pH of 9.0. Activity was significantly affected by metal ion concentrations with significantly reduced activity with $Zn^{2+}$ and $Cu^{2+}$ and increased activity with $Mg^{2+}$, $Ca^{2+}$, $Na^+$, $Ni^{2+}$ and $Co^{2+}$. Chelation using EDTA reduced activity by only 6%, suggesting metal ions are not essential for FnL activity. FnL had no significant match within the ESTHER classification system.

### Family XXII

Functional analysis of a metagenomic library produced from the rhizosphere microbiome of multiple plants identified EstD2 (53 kDa), the type protein for family XXII (*Lee et al., 2010*). EstD2 contains the GxSxG motif in the form of GHSQG. Maximum activity was identified against C4 and decreased as the fatty acid length increased. Optimum conditions were determined to be 35 °C and pH 7–9. EstD2 activity is highly sensitive to metal ions, as inhibition occurred via addition of $Mn^{2+}$, $Zn^{2+}$, $Cd^{2+}$, $Co^{2+}$, $Cu^{2+}$, $Fe^{2+}$, $Ni^{2+}$ and $Rb^{2+}$ but not $Ca^{2+}$ and $Mg^{2+}$. EDTA reduced activity by up to 68%, indicating that metal ions may be required for activity of family XXII enzymes. No similar proteins were identified within the genome database suggesting this family is present within currently uncultured bacteria, however BLAST annotation against the NCBI-NR database confirmed it as bacterial in origin. EstD2 was annotated as a member of the 'Duf_3089' family within the ESTHER system.

### Family XXIII

The type protein for family XXIII is LipA (32 kDa) and was identified within a fosmid library created from Brazilian mangrove sediment (*Couto et al., 2010*). Genome analysis identified no similarity to proteins from sequenced bacteria, however annotation against the NR database confirmed it as bacterial in origin. LipA has the pentapeptide sequence of AHSMG, fitting with the AxSxG motif. LipA was shown to have the highest activity on C10 and optimum pH and temperature of 7.5–8.5 and 35 °C respectively. LipA was annotated as belonging to the ESTHER family 'Bacterial_lipase'.

### Family XXIV

Rv0045c (35.5 kDa) is an esterase isolated from *Mycobacterium tuberculosis* (*Guo et al., 2010*). The pentapeptide GMSLG contains the catalytic serine. Maximum activity was identified against C6 and optimum conditions of 39 °C and pH 8. Genome analysis identified this family of lipolytic enzymes as specific to the *Mycobacterium*. Crystallography identified that the active site of Rv0045c is highly conserved and consists of only a serine and histidine, not the traditional catalytic triad (*Zheng et al., 2011*). The reliance of Rv0045c on the active serine has been confirmed by inactivation with PMSF (*Ortega et al., 2016*). Rv0045c was annotated to the ESTHER family 'Epoxide-hydrolase_like', however similar similarity was identified to multiple other families, suggesting it may represent its own ESTHER family.

### Family XXV

Analysis of a metagenomic library from a deep-sea sediment for lipolytic activity identified Est3L4 (~37 kDa), the type protein for family XXV (*Jeon et al., 2011*). Optimum conditions for activity were identified as 35 °C, pH 7.5 and 1.5M NaCl. The addition of NaCl consistently improved activity, suggesting this family has adapted to the saline environment of the deep-sea. Family XXV is affected by metal ions as $Mn^{2+}$, $Mg^{2+}$, $Ca^{2+}$, $Cu^{2+}$ significantly increased activity. Reduced activity was identified with addition of $Co^{2+}$ and addition of $Fe^{2+}$ inhibited all activity. EDTA was shown to have no effect, suggesting that metal ions are not essential for activity. Addition of PMSF significantly reduced activity. Est3L4 was identified as specific for long-chain fatty acids with maximum activity against C16. No similar proteins were identified within the genome database. A single transmembrane helix was detected at position 9-31. Est3L4 was annotated as a member of the 'Esterase_phb' ESTHER family.

### Family XXVI

A metagenomic library of German soil samples identified EstGS (40.7 kDa) containing the GxSxG motif in the form of GHSFG (*Nacke et al., 2011*). Maximum activity was identified against C6, with high activity also against C4. The effect of factors such as temperature, pH and metal ions were not studied. No matching proteins were identified via genome analysis but bacterial origin was confirmed via annotation against the NCBI-NR database. EstGS was identified as a member of the ESTHER family 'UCP031982'.

### Family XXVII

Alongside the discovery of family XXVII, *Nacke et al. (2011)* identified the type protein for family XXVII, EstGH (45.6 kDa), via metagenomic analysis of German soil (*Nacke et al., 2011*). The catalytic serine of EstGH was contained within GHSLG and expanded genome analysis identified similar proteins within the *Mycobacterium* which is known to exist within soil samples (*Kim et al., 2014*). EstGH had no significant match to any ESTHER family,

### Family XXVIII

The high lipolytic activity of *Rhodococcus* sp. strain CR-53, isolated from soil, led to the discovery of LipR (43 kDa), the type protein for family XXVIII (*Bassegoda, Pastor & Diaz,*

*2012*). LipR was identified to be active against medium length fatty acids with maximum activity against C10. High activity was determined on Rhodamine-triolein-supplemented agar plates. Optimum conditions were determined as 40 °C and pH 7. High concentrations of $Mg^{2+}$, $Mn^{2+}$ and $Ca^{2+}$ metal ions (10mM) were shown to increase activity whilst low concentrations (1mM) of $Fe^{2+}$, $Zn^{2+}$, $Cu^{2+}$ and $Ag^+$ significantly inhibited the lipolytic activity of LipR. The catalytic serine ($Ser^{212}$) was present within the pentapeptide GYSGG. Family XXVIII was identified to be monophyletic, occurring specifically within members of the genus *Rhodococcus*. LipR was determined to contain a single transmembrane helix from position 12-34. LipR was annotated as a member of the ESTHER family 'Fungal-Bact_LIP'.

### Family XXIX

EstGX1 (22.4 kDa) was isolated via metagenomic analysis of high Andean forest soil from Columbia (*Jiménez et al., 2012*). The catalytic serine was located within the pentapeptide GPSGG. Maximum activity was identified against C4 with optimum conditions of 40 °C and pH 8. Whilst no genomic match was identified, annotation against NR identified EstGX1 as being bacterial in origin. EstGX1 had no match to any ESTHER family.

### Family XXX

Metagenomic fosmid analysis of Chinese soil samples identified EstP2K (25 kDa) (*Ouyang et al., 2013*). Sequence analysis determined the catalytic serine was located within AHSLG and genomic analysis identified no matching sequences. Maximum activity was detected against C8 with optimum conditions of 45–55 °C and pH 7.5. Metal ions were identified to have a large effect on the activity of EstP2K with $Mg^{2+}$ doubling the activity whilst $Co^{2+}$, $Cu^{2+}$, $Zn^{2+}$, $Ca^{2+}$, $Pb^{2+}$, $Ni^{2+}$, $Mn^{2+}$, $Hg^{2+}$, $Fe^{2+}$ and $Ag^+$ all decreased the activity. Chelation using EDTA halved the enzymatic activity of EstP2K, suggesting $Mg^{2+}$ is required for activity. Activity was reduced to 1.5% with the addition of PMSF, confirming EstP2K as a serine hydrolase. EstP2K had a weak match (bitscore = 70.3) to the ESTHER family 'Lipase_2'.

### Family XXXI

Genome mining of *Psychrobacter celer* 3Pb1, a known psychrophile, identified the presence of Est12 (35 kDa) (*Wu et al., 2013*). Sequence analysis identified the catalytic serine within GHSAG and genome analysis determined this family as monophyletic to the *Psychrobacter*. Maximum activity was identified against C4 with optimum conditions of 35 °C and pH 7-8. Est12 was identified to retain activity when exposed to high concentrations of NaCl (<4.5M) for 13 h, indicating it is salt tolerant. EDTA was shown to reduce activity by 15%, suggesting metal ions are beneficial for activity but may not be essential. PMSF completely inactivated Est12 when added at 5mM. Est12 was annotated as a member of the 'Hormone-sensitive_lipase_like' family in the ESTHER system.

### Family XXXII

Est9x (32 kDa) was isolated from marine metagenomic library of the South China sea (*Fang et al., 2014*). Sequence analysis identified the catalytic serine within GHSAG and genome analysis identified highly similar proteins within the *Glaciecola*, indicating that

the original host may have been a member of this genus. PMSF significantly reduced activity, confirming Est9x as a serine hydrolase. Maximum activity was identified against C2 with decreasing activity till C14. Optimum conditions were identified as 65 °C and pH 8. Activity was removed via addition of $Co^{2+}$ whilst $Ni^{2+}$, $Cu^{2+}$, $Zn^{2+}$ all decreased activity by 50% and $Mn^{2+}$ and $Mg^{2+}$ only slightly decreased activity. The inability of EDTA to effect the activity of Est9x confirmed it as not being a metalloprotein. Est9x belongs to its own ESTHER family 'Est9X'.

### Family XXXIII

Bioinformatic analysis of a metagenomic dataset from the Solnechny hot spring in Russia led to the identification of EstDZ2 (29.4 kDa), the type protein for family XXXIII (*Zarafeta et al., 2016*). EstDZ2 activity was specific for short-chain fatty acids with maximum activity against C4. Optimum conditions were identified as 55 °C and pH 8. Metal ions and EDTA were shown to have minimal effect on activity, indicating family XXXIII are not metalloproteins. Sequence analysis identified the catalytic serine within GHSAG and an expanded search identified no similar proteins within the genome database. Activity was significantly reduced by addition of PMSF, confirming its classification as a serine hydrolase. EstDZ2 was annotated as a member of the ESTHER family 'Hormone-sensitive_lipase_like'.

### Family XXXIV

Est22 (44.56 kDa) was isolated from a metagenomic library of deep-sea sediment. Activity is specific to short length fatty acids as maximum activity was reported against C4 and no activity occurred past C10. Optimum conditions were identified as 60 °C, pH 9 and 0.5M NaCl. The inability of EDTA to alter Est22 activity suggests that metal ions are not required for activity; hence their effect was not tested. Sequence analysis identified the pentapeptide GPSMG containing the catalytic serine as well as similar proteins being present within the *Alteromonadaceae* and the expanded search identified additional matches within the *Pseudoaltermonadaceae*. PMSF was shown to significantly inhibit the activity of Est22. Est22 was identified as belonging to the ESTHER family 'Homoserine_transacetylase'.

### Family XXXV

The acetyl xylan esterase AcXE2 (25 kDa) was isolated from *Caldicellulosiruptor bescii*, a highly thermophilic species able to degrade cellulose (*Soni et al., 2017*). As an acetyl xylan esterase, AcXE2 removes the acetyl group on acetylated xylobiose and glucose. Sequence analysis identified the catalytic serine within the pentapeptide GDSIT. Whilst highly similar enzymes were only identified within the *Caldicelulosiruptor*, potential members of this family were also identified within the *Bacillaceae*, *Chthonomonadaceae* and *Paenibacillaceae*, suggesting a diverse host range. Maximum activity was identified against C2 and C4 and optimum temperature of 70 °C and pH 7.5. AcXE2 had no match to any family within the ESTHER classification system.

## CONCLUSIONS

In this paper, we have significantly expanded the classification of lipolytic enzymes to 35 families and 11 true lipase subfamilies. Defining the biochemical features of each family

along with providing type proteins allows for a greater number of potentially novel lipases to be assigned to pre-described families, preventing inconsistency within the literature. The lack of multiple members for each family with proven activity prevented comparative analysis of motifs and biochemical features.

N-terminal single transmembrane alpha-helices were detected within type proteins for four families/sub-families. Experimental evidence gained via deletion of the N-terminal transmembrane alpha-helix of LipA identified that this helix lies within the 3D structure of the protein allowing for opening of the lipase cap (*Zha et al., 2014*). It is likely that the transmembrane alpha-helices detected within the newly described families play a similar role in the structure and functionality of the proteins and do not represent membrane-bound lipases.

Previous research has shown that sequence identity does not guarantee shared functionality (*Punta & Ofran, 2008*). The average pairwise similarity between the lipolytic enzymes was 31.3% with the minimum being 14.2%. Whilst these enzymes do not share identical functionality, they have demonstrated the ability to hydrolyse the release of pNP-bound fatty acids. The low level of sequence similarity between lipolytic enzymes suggests that *in vitro* examination of proteins sharing low homology with known lipolytic enzymes may lead to identification of additional families.

Cultivation-independent investigation of environmental samples led to the identification and characterisation of multiple lipolytic families suggesting that a greater number of enzymes that have yet to be characterised likely exist. The inability of many families to be assigned to a host via genome search suggests that not only do further lipolytic enzymes remain uncharacterised, but their host species have yet to be cultured. We hope that with the recent increased effort in cultivation of novel species from environmental samples additional enzymes of biotechnological importance will be identified.

## ACKNOWLEDGEMENTS

We would like to thank Tarek Moustafa and Theresa Streidl for their invaluable feedback during the writing of this manuscript.

### Funding
Thomas Clavel received funding from the German Research Foundation (DFG) (grant no. CL481/2-1). Thomas C.A. Hitch received internal funding from the START grant program. The funders had no role in study design, data collection and analysis, decision to publish, or preparation of the manuscript.

### Grant Disclosures
The following grant information was disclosed by the authors:
German Research Foundation (DFG): CL481/2-1.
START grant program.

## Competing Interests

The authors declare there are no competing interests.

## Author Contributions

- Thomas C.A. Hitch conceived and designed the experiments, performed the experiments, analyzed the data, prepared figures and/or tables, authored or reviewed drafts of the paper, approved the final draft.
- Thomas Clavel prepared figures and/or tables, authored or reviewed drafts of the paper, approved the final draft.

## Data Availability

The code and protein sequences required for integration of additional families are available at: https://github.com/thh32/Lipase_reclassification.

## Supplemental Information

Supplemental information for this article can be found online at http://dx.doi.org/10.7717/peerj.7249#supplemental-information.

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
