# Peer review of "A proposed update for the classification and description of bacterial lipolytic enzymes"

_PeerJ, doi:10.7717/peerj.7249_

## Round 0.1 · original submission · Minor Revisions

Please address all the critical issues raised by all reviewers.

·

Basic reporting

The Manuscript tile "A proposed update for the classification and
description of bacterial lipolytic enzymes" Article ID(#37278)

Thomas C A Hitch Corresp and Thomas Clavel reporting the classification of lipolytic enzymes.-
-This study very interesting and easily understandable
-language good well written-
-figure and tables enough
-references good

Experimental design

-experimental genomic study well analysis

Validity of the findings

-over all the well presentation

Reviewer 2 ·

Basic reporting

No comment.

Experimental design

No comment.

Validity of the findings

No comment.

Additional comments

The manuscript was well-structured. The authors provided clear background for the research and defined the necessity to expand the classification system of bacterial lipolytic enzymes. With more enzymes discovered, the current system is not sufficient and various “novel” families were claimed without further analysis and confirmation. With careful research, the authors proposed an improvement of the current system based on pairwise sequence alignment, and new families and subfamilies were added. Type proteins for each of the families and subfamilies were also described with details such as conserved sequence motif and effect of metal ions. Table 1 was clear and informative, but I would suggest the authors could summarize here the consensus sequence and the metal-dependency or other representative features for easier reference.

·

Basic reporting

NO comment

Experimental design

No comment

Validity of the findings

No comment

Additional comments

The manuscript entitled “A proposed update for the classification and description of bacterial lipolytic enzymes” by Thomas Hitch and Thomas Clavel provides expanded families and subfamilies of lipolytic enzymes. At the same time, the authors also provided methods and tools to group further newly identified enzymes into the present classification system. The authors did pairwise sequence comparison to confirm or exclude the novelty of the already claimed new families. The authors not only expanded the classification system published by Kovacic et al (2019) but also corrected some errors in that system. The authors provided the most up to date and most reasonable approaches to classify the lipolytic enzymes. I only have few comments below:
1. The authors are trying to incorporate their classification system into the already published and used widely system which has been updated recently by Kovacic et al (2019). Although the authors try to be clear and not causing confusing by giving a summary section about the changes in section 3.1 Overview part, it is not clear enough. I recommend the authors only describe the expanded subfamilies/families and the families got changed in the existing system. That means in the range of family IX to XIX, the authors should only list family X, XI, XIII, and XVIII which already exist but changed by the authors.
2. In the summary table of bacterial lipolytic enzyme families, the authors need to highlight the changes they made over the existing system to make it more clear.
3. The authors should also include the bacteria origin for the enzymes listed on the table, although they did this in the text part. This is important since some of the enzymes are isolated from metagenomic libraries.
4. The protocol for origin check of metagenomics enzymes sounds very arbitrary. Can the authors provide paper evidence that this protocol has been reasonably used before or some other evidence this protocol is appreciated?

---

## Round 0.2 · accepted · Accept

All critiques were adequately addressed and the manuscript was revised accordingly. Therefore, the amended version is acceptable in its current form.

#